# Effect of Dietary Phosphate Deprivation on Red Blood Cell Parameters of Periparturient Dairy Cows

**DOI:** 10.3390/ani13030404

**Published:** 2023-01-25

**Authors:** Lianne M. van den Brink, Imke Cohrs, Lennart Golbeck, Sophia Wächter, Paul Dobbelaar, Erik Teske, Walter Grünberg

**Affiliations:** 1Department of Farm Animal Health, Utrecht University, 3584 CL Utrecht, The Netherlands; 2Clinic for Cattle, University of Veterinary Medicine Hannover, Foundation, 30173 Hanover, Germany; 3Clinic for Ruminants, Justus-Liebig-University Giessen, Frankfurter Strasse 104, 35392 Giessen, Germany; 4Department of Internal Medicine, Reproduction and Population Medicine, Faculty of Veterinary Medicine, Salisburylaan 133, 9820 Merelbeke, Belgium; 5Department of Clinical Sciences, Utrecht University, 3584 CM Utrecht, The Netherlands

**Keywords:** postparturient hemoglobinuria, hemolysis, hypophosphatemia, red blood cell

## Abstract

**Simple Summary:**

Postparturient hemoglobinuria is a rare but deadly disease of dairy cows. Cows are affected mainly in the period closely following calving. The main clinical sign—the passing of strongly discolored urine—is caused by the breakdown of red blood cells within the bloodstream. The condition is generally believed to be associated with a dietary phosphorus deficiency, but the precise etiology is unknown. We, therefore, designed two studies to explore the effect of phosphorus deprivation on red blood cell form and function. In the first study, we fed a ration low in phosphorus for four weeks before to four weeks after calving. In the second study, we restricted phosphorus only before calving. Feeding a diet low in phosphorus throughout the periparturient period (Study I), resulted in severe anemia and hemoglobinuria in a subset of cows after calving. In addition, a subclinical form of the disease was discovered. The results provide insight into, and increase awareness of, the occurrence of postparturient hemoglobinuria and the role of phosphorus.

**Abstract:**

Postparturient hemoglobinuria is a sporadic disease characterized by intravascular hemolysis and hemoglobinuria in early lactating dairy cows. The condition has empirically been associated with phosphorus (P) deficiency or hypophosphatemia; however, the exact etiology remains obscure. This paper summarizes two controlled studies investigating the effect of P deprivation during the transition period. In Study I, 36 late pregnant dairy cows were randomly assigned to either a diet with low, or adequate, P content from four weeks before calving to four weeks after calving. In Study II, 30 late pregnant dairy cows were again assigned to either a diet with low, or adequate, P for the last four weeks before calving only. Pronounced hypophosphatemia developed during periods of restricted P supply. In early lactation, a subtle decline of the red blood cell count occurred independently of the dietary P supply. In Study I, anemia developed in 11 cows on deficient P supply, which was associated with hemoglobinuria in five cases. Neither erythrocyte total P content nor osmotic resistance of erythrocytes were altered by dietary P deprivation. Restricted dietary P supply, particularly in early lactation, may lead to postparturient hemoglobinuria, but more frequently causes clinically inapparent hemolysis and anemia in cows.

## 1. Introduction

The passing of strongly discolored urine, with colors ranging from red to black, is the main clinical sign of postparturient hemoglobinuria (PPH). This sporadic disease of multiparous, high-producing dairy cattle is characterized by intravascular hemolysis, hemoglobinuria, and anemia [1]. The occurrence of the disease is concentrated around the first two to four weeks of lactation and, although PPH is a sporadic disease, it has concurrent high mortality [2,3]. Research on this disease is mostly performed in field cases, primarily because attempts to experimentally induce PPH by feeding rations low in phosphorus (P) were successful incidentally only [4,5].

Low plasma concentrations of inorganic phosphorus (Pi) have been found in acutely affected animals [2,3,6]. Therefore, mechanisms linking hypophosphatemia and red blood cell (RBC) destruction have been postulated. A decline of adenosine triphosphate (ATP) in the RBCs, caused by reduced availability of intracellular Pi, has been proposed as the most important causative factor of this condition [7]. This hypothesis, however, does not provide a satisfactory explanation for the occurrence of PPH in cows with normal plasma [Pi], nor the meagre response to treatment with phosphate salts repeatedly reported in the literature [8,9,10]. Furthermore, the discrepancy between the substantial incidence of hypophosphatemia in dairy cattle and the very low incidence of PPH remains unresolved [11]. Other causative or contributing factors, such as oxidative stress, copper (Cu) deficiency, or ingestion of hemolytic saponins have also been discussed [1,10,12], as have changes in the structure and morphology of the RBC membrane, caused by a decrease in total phospholipids in states of P deficiency [13].

Even in healthy transitioning dairy cows, a decrease of PCV over the first few weeks of lactation is frequently found [14,15,16]. Hemoconcentration in late gestation and around parturition, and ensuing reversal hemodilution in the days following parturition, was brought forward as an explanation for observed changes in PCV during early lactation in dairy cows. The increase in plasma volume (i.e., hemodilution) is considered a homeostatic mechanism to maintain adequate fluid supply to the mammary glands for milk production [17,18]. The question remains whether an additional loss of RBCs might also occur in the first weeks of lactation.

The objective of the two controlled studies presented here was to obtain a better understanding of the etiopathogenesis of this condition and, in particular, to elucidate the role of phosphorus. More specifically, the aim was to study the effects of dietary P deprivation on RBC parameters in periparturient dairy cows not only at a clinical, but, more importantly, at a subclinical level under strictly controlled conditions.

## 2. Materials and Methods

The present report communicates the results of 2 distinct studies investigating the effect of dietary P deprivation either throughout the transition period, or only during late pregnancy, on the erythron of dairy cows. These studies form part of a multi-institutional project investigating the effect of P deprivation on various tissues and organ systems in transition dairy cows. The results of this project have been published by different research groups [19,20,21]. Ethical Statement: The national and institutional guidelines for the care and use of experimental animals were followed and all experimental procedures were approved by local Animal Care and Use Committee (Study I: DEC; permit no. AVD108002016616, Study II: Animal Welfare and Ethics Committee of the government of Coblenz, Rhineland Palatinate, Coblenz, Germany, permit no. 23-177-07/G 19-20-008). 

### 2.1. Study I. P Deprivation in Transition Cows

#### 2.1.1. Study Design, Animal Housing and Feeding

This study investigated the effect of dietary P deprivation from 4 weeks before to 4 weeks after calving in dairy cows. The study has been described in detail elsewhere [19]. Briefly, this randomized, controlled and duplicated study included a total of 36 healthy, multiparous, pregnant Holstein–Friesian (HF) or HF crossbreed dairy cows. The study was conducted in 2 consecutive replicates with 18 cows each. Cows were housed in individual tie stalls with rubber bedding covered with sawdust in a temperature-controlled facility. 

In each replicate, 18 cows expected to calve within the same week were blocked by lactation number (LN) and, within each block, paired by milk yield of the previous lactation. The pairs were then randomly distributed between the two treatments which were either a P deficient diet (LP) or a diet with adequate P content (AP). The study started 6 weeks before the expected calving date with a 2-week acclimation period during which cows of both treatments were fed the dry cow diet with adequate P content (AP diet). Subsequently, LP cows were switched to the LP diet, while AP cows remained on the diet with adequate P content. Experimental diets were fed from 4 weeks before the expected week of calving until 4 weeks post-partum (p.p.). 

The determined P content in dry and lactating cow diets were respectively 0.15% and 0.20% in dry matter for the LP treatment and 0.28% and 0.44% for the AP treatment. Details of the experimental diets have been published previously (https://doi.org/10.1371/journal.pone.0219546.s003, accessed on 23 December 2022). Cows of the first replicate were enrolled in a repletion period at the end of the depletion phase, during which cows of both treatments were offered the same AP lactating cow ration. Access to feed was restricted to 12.5 kg DM/d for dry cows and was ad libitum for lactating cows, and cows had ad libitum access to water.

#### 2.1.2. Data and Sample Collection

Data on animal health, feed intake and milk yield were collected and processed as previously described [19]. 

Blood was drawn by venipuncture of a jugular vein into blood tubes containing EDTA as anticoagulant (EDTA Vacuette®, Greiner Bio-One, Kremsmünster, Austria) and into blood tubes containing lithium–heparin as anticoagulant (Li–heparin Vacuette®, Greiner Bio-One, Kremsmünster, Austria) between 0700 h and 0800 h. Blood samples of study cows were obtained weekly and analyzed for a variety of parameters that we published previously [20]. Blood samples included in the database of the present study were obtained at different sampling times (STs) throughout the study, depicted in Figure 1. ST1 at the end of acclimation, ST2 after 2 weeks of P depletion, ST3 within 1 week of parturition, ST3.5 at 18 ± 4 d after parturition (replicate 2 only), ST4 at the end of the depletion period, and ST5 at the end of P repletion (replicate 1 only). 

The additional sampling time ST3.5 was implemented in the second replicate, as a result of the observation that PPH occurred between ST3 and ST4 in the first replicate. Blood samples were, furthermore, obtained from cows affected by clinical PPH during the hemolytic period. Heparinized blood was centrifuged within 30 min of collection at 1000× *g* for 15 min at 6 °C. Harvested plasma was stored at −21 °C until analyzed, as described below. 

Liver biopsies were performed for liver Cu determination. Samples were collected at the sampling times ST1 to ST5, and processed as previously described [20]. 

Urine was grossly checked for discoloration suggestive of hemoglobinuria at least every other day from the day of calving. Urine samples were collected once weekly between 0600 h and 0700 h from spontaneous urination. Urine was stored at −21 °C until analyzed as described below. 

#### 2.1.3. Sample Processing and Analyses

Plasma [Pi] (ammonium molybdate method), non-esterified fatty acids ([NEFA]; acetyl-CoA-synthetase–acetyl-CoA-oxydase method), total bilirubin ([TBil], dichloraniline method) were determined on an automated analyzer (ABX Pentra 400; Horiba, Europe GmbH, Langenhagen, Germany).

EDTA blood was used for RBC counts as well as for determination of packed cell volume (PCV), mean corpuscular volume (MCV) and mean corpuscular hemoglobin (MCH). All measurements were conducted within 1 h of blood collection on an automated haematology system containing multispecies software (Advia 2120i, Siemens Healthcare Diagnostics GmbH, Eschborn, Germany).

RBC osmotic resistance was studied in all cows of the first replicate as reported previously [11,22] within 1 h of blood collection. In short, 50 μL of heparinized blood were added to each of 14 vials containing 5 mL sodium chloride (NaCl) solutions spanning a concentration range from 0.1 to 0.9%. Vials were then incubated at room temperature for 30 min before centrifuging for 5 min at 1200× *g*. The degree of hemolysis was determined spectrophotometrically by measuring the absorbance at a wavelength of 540 nm in the supernatant. Spectrophotometric absorbance of blank isotonic NaCl solution (0.9%) was used as equivalent to 0% hemolysis and blood mixed with 0.1% salt solution was used as equivalent to 100% hemolysis. The concentrations of the salt solutions were then plotted against the degree of hemolysis (in percent) and the concentration of the NaCl solution at which 10% and 90% hemolysis occurred were obtained from the plotted curve. Osmotic resistance of each sample was, therefore, characterized by the concentration of the NaCl solution at which 10% (OSM10) and 90% (OSM90) of hemolysis occurred.

For the determination of intracellular P, potassium (K) and sodium (Na) content of RBCs, cells were prepared as reported previously [11,22]. Briefly, Li–Heparin tubes were kept at room temperature and centrifuged within 1 h of sample collection at 1600× *g* for 10 min. The remaining packed cells were washed 3 times by adding 1 part of isotonic NaCl to 1 part of packed cell volume. One mL of washed packed cells was then added to 5 mL of deionized water to induce hemolysis. The lysate was then centrifuged for 10 min at 1600× *g* and the supernatant collected for biochemical analysis. Total intracellular P (P-RBC), K (K-RBC) and Na (Na-RBC) content of hemolyzed RBC was determined by ICP–OES.

Liver Cu content, considered the most accurate parameter to diagnose Cu deficiency, was determined to assess the Cu status of the study animals [23]. Liver biopsy specimens were processed, as described elsewhere [20]. The liver Cu content was determined by ICP–OES. 

Urine samples of replicate 1 were analyzed for creatinine (enzymatic method) and Na (Ion selective electrode) on an automated analyzer (ABX Pentra 400; Horiba, Europe GmbH, Langenhagen, Germany) in order to calculate Na:creatine ratios. Discolored urine was spun to rule out hematuria.

#### 2.1.4. Data Analysis

Based upon the occurrence of PPH in the first 4 weeks of lactation and the development of the PCV during that time period, LP cows were retrospectively assigned to 1 of 3 categories of hemolysis that were LP-PPH0, LP-PPH1 or LP-PPH2. Cows having developed clinical signs of PPH (i.e., hemoglobinuria) were categorized as LP-PPH2; cows with a decline of PCV between ST3 and ST4 of over 20%, which resulted in a PCV below 0.22 L/L at ST4, were categorized as LP-PPH1, and LP cows of which the PCV showed a decline of less than 20% between ST3 and ST4, were classified as LP-PPH0. Since animals in LP-PPH1 did not display hemoglobinuria and remained clinically healthy, this group was also referred to as subclinical PPH. The threshold for the PCV of 0.22 L/L was chosen as this corresponded to the mean PCV− 2 SD at ST4 in AP cows. Based on the criteria mentioned above, all AP cows would have been categorized as PPH0. By design, classification into PPH categories only included LP cows, thus, representing sub-groups of the LP treatment.

### 2.2. Study II. P Deprivation in the Dry Period

#### 2.2.1. Study Design, Animal Housing and Feeding

The second study investigated the effect of dietary P deprivation during the last 4 weeks of the dry period in dairy cows. The study has been described in detail elsewhere [21]. Briefly, thirty late pregnant multiparous HF cows on a research farm, that were housed in a free stall barn, were used for this randomized and controlled study. 

Cows enrolled in the study were randomly allocated to 1 of 2 treatments that was either a diet with low P (LPAP: low P antepartum) or with adequate P content (APAP: adequate P antepartum) offered from 4 weeks before expected calving to the moment of calving. The experimental feeding period was preceded by a 2-week acclimation period, during which all cows were offered the APAP dry cow diet with adequate P content. Subsequently, LPAP cows were switched to the LPAP diet for the remainder of the dry period, while APAP cows remained on their diet with adequate P content. After calving, all cows received a diet formulated for lactating cows with adequate P content. The study period extended from 6 weeks before the expected calving date to the 6th week of lactation. The diets contained 0.16% and 0.30% P in DM for LPAP and APAP, respectively. The lactating cow ration contained 0.46% P in DM. A table summarizing the composition of the experimental diets has been provided (Appendix A, [21]). Access to feed was restricted to 11.5 kg of DM/d for dry cows and was ad libitum for lactating cows, and cows had ad libitum access to water. 

#### 2.2.2. Data and Sample Collection

Animals were monitored daily. Cows were milked twice daily between 0430 h and 0530 h and between 1530 h and 1630 h. Milk yields of each milking were recorded automatically. 

Blood was drawn once a week at standardized times between 0800 h and 1000 h by venipuncture of a jugular vein and collected in blood tubes containing EDTA (EDTA Vacuette®, Greiner Bio-One, Kremsmünster, Austria) and tubes containing Li-heparin (LH Vacuette®, Greiner Bio-One, Kremsmünster, Austria) as anticoagulant. 

#### 2.2.3. Sample Processing and Analysis

Li-heparin blood tubes were centrifuged within 20 min of collection at 1730× *g* for 15 min at room temperature. Harvested plasma was stored at −21 °C until analyzed for plasma Pi, NEFA and TBil as described above. 

Blood collected in EDTA tubes was shipped to a diagnostic laboratory at room temperature for determination of the RBC counts and PCV, MCV and MCH on an automated haematology system with multispecies software (Advia 2120i, Siemens Healthcare Diagnostics GmbH, Eschborn, Germany) within 24 h of sample collection.

### 2.3. Statistical Analyses

Data are presented as least square means (LSM) ± SEM, or median and 95% CI for data not meeting the requirement of normality of residuals and homogeneity of variance. Values were log-transformed when necessary to achieve normal distribution. In addition, data on individual cows during the clinical phase of disease and their recovery are provided separately. Repeated measured analyses of variance were used. The most appropriate covariance structure was chosen, based on the lowest Akaike information criterion. Repeated measures mixed model regression analyses were conducted using PROC MIXED (SAS 9.4, SAS Inc., Cary, NC, USA). 

For data of Study I, several models were constructed. A first model studied effects of treatment (AP vs. LP), replicate, time, LN, and the interactions of treatment x time. This model did not take into consideration the occurrence of PPH. Animal ID was defined as subject, and time was the repeated factor. The same model only including LP cows without signs of hemolysis (LP-PPH0) was constructed to identify differences between AP cows and LP-PPH0 cows. A third model only including LP cows was constructed to test for the effect of PPH status (group), replicate, time, LN, and the interactions between group x time in LP cows. LN and replicate were removed from the models when found to be insignificant. The significance level was set at *p* < 0.05. Bonferroni-adjusted P values were used to assess differences between time points and between PPH groups whenever the F test was significant. 

For data of Study II models, to determine effects of treatment, time, LN, and the interactions between treatment x time were constructed. LN was removed from the final model when found to be insignificant. Animal ID was defined as subject, and time was the repeated factor. The significance level was set at *p* < 0.05. Bonferroni-adjusted P values were used to assess differences between time points whenever the F test was significant. 

Pearson and Spearman correlation analyses were conducted to determine associations between parameters related to the erythron with blood biochemical parameters. These analyses were conducted stratified by week relative to calving, and covered the period from 4 weeks before calving (week −4) to 6 weeks after calving (week +6).

The sample sizes of both studies was based on power analyses, that were based on treatment effects and variation of outcome variables determined for other parts of this project [19,24]. The occurrence of PPH, however, was unpredictable and a prospective power analysis for the data presented here was, therefore, not possible.

## 3. Results

### 3.1. Study I. P Deprivation in Transition Cows

Of the 36 enrolled cows, four cows—all animals assigned to the LP treatment—were prematurely released from the study between the 11th and 13th day of lactation; thus, between the time points equivalent to ST3 and ST3.5. One animal was removed after having been erroneously offered the AP diet (12 d p.p.). Three animals were eliminated between 11th and 13th d p.p. because of severe PPH requiring blood transfusion. In total, clinical signs of PPH in the form of gross discoloration of urine developed in five of the 18 LP cows. The remaining two cows with clinical PPH (LP-PPH2) completed the study after having made a partial recovery without therapeutic intervention (see Section 3.1.7. Clinical Cases). Of the LP cows, in total seven animals were categorized as LP-PPH0, six cows as LP-PPH1 and five as LP-PPH2. Due to the low number of LP-PPH2 animals remaining in the study after ST3 (n = 2), blood biochemical and hematological parameters of this group are presented for each animal stratified by sampling time in Table 1 (also see Section 3.1.7. Clinical Cases). The number of animals in each LP-PPH subgroup was not even across replicates. Replicate 1: LP-PPH0 (n = 5), LP-PPH1 (n = 2) and LP-PPH2 (n = 2). Replicate 2: LP-PPH0 (n = 2), LP-PPH1 (n = 4) and LP-PPH2 (n = 3).

#### 3.1.1. Plasma Biochemical Parameters

The development of plasma [Pi] over time, stratified by treatment and LP-PPH group, is presented in Figure 2 and Table 1. The analysis of variance (ANOVA) included all cows with treatment as fixed effect revealed treatment- (*p* < 0.0001), time- (*p* < 0.0001) and treatment x time interaction (*p* < 0.0001) effects. A significant transient decline of [Pi] at ST3, compared to ST1, ST2 and ST3.5, was observed for AP cows, although values for these animals otherwise remained within the reference range throughout the study (1.4–2.3 mmol/L, [25]). In LP cows, a steep decline in plasma [Pi] occurred between ST1 and ST2, and values further decreased until nadir was reached and maintained between ST3 and ST4. For replicate 1, in which experimental cows also underwent a P repletion phase, this was followed by a steep increase returning to baseline values between ST4 and ST5 (Figure 2). Throughout the P depletion phase (between ST2 and ST4) the plasma [Pi] levels of LP cows were significantly lower than AP cows, and were below the reference range for cattle [25]. When repeating the ANOVA only including AP and LP-PPH0 cows, treatment (*p* < 0.0001), time (*p* < 0.0001) and treatment x time interaction (*p* < 0.0001) effects were found, resembling the results described above. The ANOVA for LP cows with PPH as fixed effect (group) also showed time (*p* < 0.0001) effects, but no group or group x time interaction effects.

The development of TBil over time, stratified by treatment and LP-PPH group, is presented in Figure 3 and Table 1. The ANOVA for all cows with treatment as fixed effect revealed treatment (*p* = 0.05), time (*p* < 0.0001), LN (*p* = 0.02), and treatment x time interaction (*p* < 0.02) effects. Overall estimates for TBil for LN 2, 3, 4 and 5 were, respectively, 4.9 ± 0.4; 4.6 ± 0.4; 7.2 ± 0.8 and 7.2 ± 1.2 μmol/L. The TBil value of LN 4 was significantly elevated, compared to LN 2 and LN 3 (*p* < 0.05). The TBil of AP and LP both peaked at ST3, with LP cows reaching higher values. Concentrations returned to antepartum values by ST3.5 in both treatments. In contrast to LP cows, the TBil of AP remained within the reference range for cattle throughout the study (<8.6 μmol/L; [26]. The ANOVA only including AP and LP-PPH0 cows identified time (*p* < 0.0001), and LN (*p* = 0.04), but no treatment or treatment x time interaction effects (Figure 3). The ANOVA for LP cows with PPH as fixed effect (group) showed time (*p* < 0.0001) effects, but no group or group × time interaction effects. LP-PPH1 showed significantly higher values at ST3 compared to preceding and successive STs. The observed time effects of LP-PPH0 and LP-PPH2 did not reach significance level.

#### 3.1.2. Erythron-Related Parameters

The RBC count–time curves of the AP and LP-PPH groups are presented in Figure 4 and Table 1. PCV data of the same groups are provided in Table 2. The analysis of variance for all cows with treatment as fixed effect revealed treatment (*p* = 0.002; *p* = 0.01), time (*p* < 0.0001; *p* < 0.0001) and treatment x time interaction (*p* < 0.0001; *p* = 0.002) effects on RBC count and PCV, respectively. RBC and PCV of both treatments significantly decreased in early lactation, with a more pronounced decrease in LP, compared to AP animals. The ANOVA only including AP and LP-PPH0 cows showed time (*p* < 0.0001; *p* < 0.0001), but no treatment or treatment x time interaction effects for RBC or PCV, respectively. In both AP and LP-PPH0 RBC count decreased significantly between ST3 and ST4 (Figure 4). Nevertheless, values remained within the reference range (5.1–7.6 × 10^12^/L, [26]. A similar pattern was found for PCV (Table 2; 22–33% [26]). The ANOVA for the LP cows with PPH as fixed effect (group) showed group (*p* = 0.0005; *p* < 0.0001), time (*p* < 0.0001; *p* < 0.0001) and group × time interaction (*p* < 0.0001; *p* = 0.0002) effects on RBC count and PCV, respectively. The RBC counts of LP cows were higher in the first replicate, compared to the second (respectively, 5.7 ± 0.1 and 5.2 ± 0.1 × 10^12^/L, *p* = 0.04). The stratified analysis revealed significant differences between the PPH groups after parturition (Figure 4; Table 2). From ST3.5 on, the RBC counts of both LP-PPH1 and LP-PPH2 were significantly lower than LP-PPH0. The values fell below the reference range, with LP-PPH2 dropping significantly lower than LP-PPH1 at ST3.5 and ST4. PCV of LP-PPH1 and LP-PPH2 were below the reference range at ST3.5 and ST4, but no longer so at ST5. 

The development of the MCV over time of AP and LP-PPH cows is presented in Figure 5 and Table 1. The ANOVA including all cows with treatment as fixed effect revealed treatment (*p* = 0.0008), replicate- (*p* < 0.0001), time (*p* < 0.0001), and treatment x time interaction (*p* < 0.0001) effects. Overall, estimates of replicate 1 (44.2 ± 0.7 fL) were significantly lower than replicate 2 (48.7 ± 0.7 fL). The MCV of AP cows did not change over time, whereas the MCV of LP cows rose, from ST3.5 till the end of the study. Time (*p* < 0.0001) and replicate (*p* = 0.0003) effects, but no treatment or treatment x time interaction were identified for MCV when repeating the ANOVA only including AP and LP-PPH0 cows. The ANOVA for LP cows with PPH as fixed effect (group) showed no group effect, but replicate (*p* = 0.003), time (*p* < 00001), and group x time interaction (*p* < 00001) effects. Overall, estimates of replicate 1 (46.5 ± 1.0 fl) were significantly lower than replicate 2 (51.5 ± 1.0 fl). In contrast to LP-PPH0, MCV of LP-PPH1 and LP-PPH2 rose significantly after parturition (Figure 5, Table 1) exceeding the reference range for cattle (38–50 fL, [26]). For LP-PPH1 this rise was already significant at ST3, after which it further increased and remained elevated till the end of the study. The MCV of LP-PPH2 was significantly elevated at ST3.5, compared to preceding samplings, and exceeded LP-PPH1 values at ST4. At ST5, although significantly lower than ST4, values were still elevated, compared to ST1-ST3.5 (Table 1).

The development of the MCH over time of AP, LP and LP-PPH groups are presented in Table 3. The ANOVA with all cows including treatment as fixed effect revealed treatment (*p* = 0.003), replicate (*p* = 0.002), time (*p* < 0.0001) and treatment x time interaction (*p* < 0.0001) effects. Overall, estimates of replicate 1 (1.03 ± 0.01 fmol) were significantly lower than those for replicate 2 (1.10 ± 0.01 fmol). The MCH of AP did not change over time, whereas the MCH of LP was elevated between ST2 and ST3, compared to ST1. Then, values further increased, and remained significantly higher than AP, from ST3.5 till the end of the study. When repeating the ANOVA only including AP and LP-PPH0 cows, time (*p* < 0.0001) and replicate (*p* = 0.002), but no treatment or treatment x time interaction effects, were identified for MCH. The MCH of AP and LP-PPH0 remained within reference range throughout the study (0.87—1.12 fmol, [26]). The ANOVA for LP cows with PPH as fixed effect (group) showed no effect of group, but time (*p* < 0.0001) and group x time interaction (*p* < 0.0001) effects were exhibited. In contrast to LP-PPH0, MCH of LP-PPH1 and LP-PPH2 significantly increased and remained elevated above the reference range from ST3.5 till the end of the study (Table 3). The MCH of LP-PPH2 was significantly higher than LP-PPH1 at ST4. At ST5, the MCH of LP-PPH1 and LP-PPH2 did not differ, but were still elevated compared to LP-PPH0.

#### 3.1.3. Osmotic Fragility

The results of the osmotic fragility test, stratified by treatment and LP-PPH group, are presented in Figure 6. The ANOVA with all cows and treatment as fixed effect revealed no time or treatment x time interaction effects for both OSM10 and OSM90, but effect of treatment (*p* = 0.009; *p* = 0.004, respectively). Hemolysis at the 10% level occurred at higher NaCl concentrations (higher tonicity) in LP cows, compared to AP cows at all time points of the study (LP: 0.64 ± 0.01; AP: 0.60 ± 0.01 % NaCl). The same effect was observed for 90% hemolysis (LP: 0.49 ± 0.01; AP: 0.46 ± 0.01 % NaCl), indicating an increased osmotic fragility.

The ANOVA only including AP and LP-PPH0 cows revealed treatment- (*p* = 0.04) but no time or treatment x time interaction effects for OSM10. No treatment, time, or treatment x time interaction effects were identified for OSM90. Concentrations at which 10% hemolysis occurred were significantly higher in LP-PPH0 between ST3 and ST5, compared to AP. 

The ANOVA for LP cows with PPH as fixed effect (group) showed no group, but time (*p* = 0.009) and group x time interaction (*p* = 0.006) effects. None of these effects were present for OSM90. A stratified analysis by PPH status revealed only numerical differences between the groups. The time effect of osmotic fragility did not reach significance level for LP-PPH0 and LP-PPH1 throughout the study. For OSM10, the LP-PPH2 cows had significantly lower values at ST4 and ST5, compared to the preceding sampling times (Table 1).

#### 3.1.4. Intracellular Electrolyte Content of RBC

The concentration–time curves for P–RBC stratified by treatment and PPH group are presented in Figure 7A. Data of the LP-PPH2 cows during clinical disease and recovery are provided separately in Table 1. The ANOVA for all cows with treatment as fixed effect revealed no effect of treatment, time, or treatment x time interaction effects. The ANOVA only including AP and LP-PPH0 cows yielded the same results. The ANOVA for LP cows with PPH as fixed effect (group) also did not show treatment, time, or treatment x time interaction effects. None of the changes over time, or differences between any of the groups, reached significance level (Figure 7A). 

The concentration–time curves for K–RBC stratified by treatment and PPH group are also depicted in Figure 7B. The ANOVA for all cows with treatment as fixed effect revealed no treatment or time effects, but did reveal a treatment x time interaction (*p* = 0.03) effect. At ST4, K–RBC of LP cows was significantly higher than AP cows. The ANOVA only including AP and LP-PPH0 cows did not identify treatment, time or treatment x time interaction effects. The ANOVA for LP cows with PPH as fixed effect (group) showed no group effect, but time (*p* < 0.0001) and treatment x time interaction (*p* < 0.0001) effects. A statistically significant elevation of K–RBC was present in LP-PPH2 cows at ST4 and ST5, with values exceeding LP-PPH0 and LP-PPH1 (Table 1). In LP-PPH0 cows a small, but statistically significant decrease of K–RBC occurred between ST4 and ST5. Observed changes over time in K–RBC of LP-PPH1 did not reach a significant level.

The concentration-time–curves for Na–RBC, stratified by treatment and PPH group, are presented in Figure 7C. The ANOVA including all cows with treatment as fixed effect showed no effect of treatment, but did reveal time (*p* = 0.001), LN- (*p* = 0.03), and treatment x time interaction (*p* = 0.02) effects. Overall, estimates for Na–RBC for LN 2, 3 and 4 were respectively 109 ± 1.8; 117 ± 2.1; 107.9 ± 3.6 mmol/L. Differences between treatments were only observed at ST1, with higher values in AP cows compared to LP. Within treatments, AP cows had higher Na–RBC at ST1 compared to ST2 and ST5 and a significant decline was observed in LP cows between ST4 and ST5. The ANOVA only including AP and LP-PPH0 cows revealed time (*p* = 0.02), but no treatment or treatment x time interaction effects. The ANOVA for LP cows with PPH as fixed effect (group) showed group (*p* = 0.02), time (*p* < 0.01), and group x time interaction (*p* < 0.02) effects. Changes over time in Na–RBC of LP-PPH0 and LP-PPH1 did not reach significance level. At ST4, Na–RBC was higher in LP-PPH2 cows, compared with other sampling times, and compared to LP-PPH0 and LP-PPH1 (Table 1). These differences also existed at ST5, between LP-PPH2 and LP-PPH0.

#### 3.1.5. Liver Cu, Urine, Milk Production and Feed-intake

The median and 95% CI for liver Cu content, stratified by treatment, are presented in Table 4. 

The ANOVA for all cows with treatment as fixed effect revealed no treatment effect, but revealed time (*p* < 0.0001) and treatment x time interaction (*p* = 0.0005) effects. A transient decline of liver Cu occurred at ST4 in AP cows. Only at ST5, did liver Cu differ significantly between AP and LP cows, with the latter having a lower Cu content. The ANOVA only including AP and LP-PPH0 cows revealed no treatment or time effects, but treatment x time interaction (*p* = 0.0003) effects were evident. Stratified analysis did not reveal differences between AP and LP-PPH0 cows at any of the STs. The ANOVA for LP cows with PPH group as fixed effect (group) showed no group, time, or group x time interaction effects. 

The urine Na:creatinine ratio over time of AP and LP cows are presented in Figure 8. The ANOVA for all cows with treatment as fixed effect revealed effects of treatment- (*p* = 0.003) and time (*p* < 0.0001), but no treatment x time interaction effect. Overall, Na:creatinine ratio in urine was lower in LP cows, compared to AP cows (0.5 ± 0.1 and 0.8 ± 0.1, respectively).

Urine [Na:creatinine] decreased in early lactation in both treatments. The urine Na:creatinine ratio was below antepartum (week -1) values from week 1 to 5 p.p. in LP cows and from week 2 to 4 in AP cows. The ANOVA only including AP and LP-PPH0 cows, revealed effects of treatment (*p* = 0.003) and time (*p* = 0.0005), but no treatment x time interaction effects, resembling the results described above. The ANOVA for LP cows with PPH as fixed effect (group) also showed time (*p* < 0.0001) effects, but no group or group x time interaction effects, resembling the LP results described above.

#### 3.1.6. Correlation Analysis

Spearman correlation analyses investigating the associations between parameters related to the erythron and blood biochemical parameters stratified by sampling time at ST3 revealed negative associations of plasma [Pi] with OSM10 (rho = −0.53, *p* = 0.02) and OSM90 (rho = −0.55, *p* = 0.02) and of P–RBC with OSM10 (rho = −0.48, *p* = 0.04) and OSM90 (rho = −0.66, *p* = 0.003). For ST 3.5 RBC was associated with the RBC (rho = 0.66, *p* = 0.005), plasma NEFA (rho= −0.54, P=0.03), TBil (rho = −0.71 *p* = 0.002) and GGT (rho = −0.59, *p* = 0.02) At ST-4 Plasma [Pi] was found to be associated with RBC (rho = 0.48, *p* = 0.005) and OSM90 (rho = −0.58, *p* = 0.01).

#### 3.1.7. Clinical Cases

Cow 3739, entering her third lactation, was clinically unapparent until nine days p.p. when she developed hemoglobinuria with dark brown urine. At that time, her PCV was 0.21 L/L and her plasma showed discoloration indicative of hemolysis. Hematologic and plasma biochemical results from the time of calving are presented in Table 1. After blood sampling on day 11 p.p., and, thus, after three days of overt hemoglobinuria the cow was excluded from the original study and switched to the AP ration. The animal was furthermore treated twice daily with 250 g NaH_2_PO_4_ orally, and the course of the disease was further followed. The treatment resulted in marked increase of plasma [Pi] within hours, while the decline of the PCV and hemoglobinuria persisted and the cow deteriorated clinically. On the peripheral blood smear on day 13 numerous spherocytes, and a high degree of anisocytosis, polychromasia and some basophilic stippling were noted. Immediately after blood sampling on the morning of day 13 p.p. the animal received the first of two 5 L blood transfusions, one day apart, and made an uneventful full recovery thereafter. 

Cow 8183 entering her fourth lactation developed gross hemoglobinuria on day 14 after calving. The results of blood samples taken on subsequent days and adjacent STs are presented in Table 1. This animal was also followed intensely and remained in the study and on the LP ration, without any treatment or supplementation as the PCV did not drop below 0.15 L/L, and the color of her urine normalized by day 18 p.p.

The results of osmotic fragility determination of both cows during the clinical phase of the disease are given in Figure 9A. For cow 3739 an increase in osmotic fragility was observed during the clinical phase of the disease, and 10% hemolysis occurred in 0.9% NaCl solutions. A transient increase in osmotic fragility was also present in cow 8183. At this time, RBC counts had dropped by more than half, compared to antepartum values.

Intracellular RBC P, K and Na of these two cows are presented in Table 1. Compared to the data of the other treatments and PPH-subgroups, low K–RBC and high Na–RBC concentrations were found in the LP-PPH2 cows during clinical disease. An overt increase in K–RBC and Na–RBC were noted in cow 8183 at ST4 and ST5. At these timepoints, OSM10 also had lower values, compared to preceding sampling times (Figure 9B).

During replicate two, another three cows, all entering their fourth lactation developed PPH (i.e., LP-PPH2) between 10–11 days p.p. Severe anemia and hyperbilirubinemia were present. Two cows (1261 and 3340) were both eliminated from the study two and three days after onset of hemoglobinuria and received blood transfusions as the PCV reached 0.12 L/L and urine color did not ameliorate. One cow (5035) remained in the study untreated and showed resolution of hemoglobinuria and intravascular hemolysis within four days after onset of clinical signs. PCV, RBC, Plasma Pi, TBil and MCV values on days preceding and during the clinical disease are presented in Table 1 also.

### 3.2. Study II. P Deprivation in the Dry Period

In the second study, in which cows assigned to the LPAP treatment were subject to dietary P deprivation only for the last four weeks of pregnancy, but not for the early lactation period, none of the studied cows developed signs of clinical or subclinical PPH. One LPAP cow was prematurely removed from the study in the first week of lactation after having developed signs of acute rumen acidosis. The present report communicates the RBC parameter fluctuations in the periparturient period of these cows. The plasma [Pi]–time curves, TBil–time curve and data on milk yield were published previously [21].

#### 3.2.1. Erythron-Related Parameters

The development of PCV over time, stratified by treatment, is presented in Figure 10. For the PCV an effect of time (*p* < 0.0001), but no treatment or treatment x time interaction effects were observed. The PCV remained constant in the weeks preceding parturition. After parturition PCV decreased over the first four weeks of lactation, after which PCV remained constant till the end of the study. 

Similarly, an effect of time (*p* < 0.0001), but no treatment or treatment x time interaction effects were identified for the RBC. RBC counts of the APAP and LPAP cows are presented in Table 5. As with the PCV, the RBC counts remained constant prior to parturition, increased slightly in the first week of lactation, but then steadily declined till the end of the study. For both groups PCV and RBC values remained within reference range for cattle throughout the study.

The MCV data of APAP and LPAP cows are presented in Table 5. The MCV revealed an effect of time (*p* < 0.0001), but no effect of treatment or treatment x time interaction effects. MCV values were highest close to parturition. 

The MCH of APAP and LPAP cows are presented in Table 5. For the MCH no treatment, time or treatment x time interaction effects were observed. 

#### 3.2.2. Correlation Analysis

The spearman correlation analysis of parameters related to the erythron and blood biochemical parameters of weekly blood samples stratified by week relative to calving, only revealed a negative association between RBC and TBil at week +2 (rho = −0.54, *p* = 0.002). While no association was identified between RBC and NEFA, there was a positive correlation between TBil and NEFA (rho = 0.53, *p* = 0.002). The same analysis conducted, stratified by treatment, showed that TBil was negatively associated with RBC at week +2 in LP (rho= −0.53, *p* = 0.05), while this effect did not reach significance level in AP cows (rho= −0.47, *p* = 0.07).

## 4. Discussion

The studies presented here report the effect of dietary P deprivation either only during the last four weeks of gestation until calving or during the transition period extending from the last four weeks of gestation until four weeks after calving. To the best of our knowledge, Study I is the first randomized and controlled study that reports multiple cases of experimentally-induced PPH in cows on dietary P deprivation until the fourth week of lactation. 

This study, furthermore, identified a group of cows showing a marked decline in the PCV and RBC count in the first weeks of lactation that was not associated with overt hemoglobinuria and only occurred in P-deprived animals. This finding suggests the existence of a subclinical presentation of PPH associated with P deprivation, which had not been noticed before [5]. Remarkably, PPH at a clinical and subclinical level only occurred when P deprivation was extended into early lactation. 

The subdivision of the LP group into subgroups obviously led to low statistical power, due to the small sample size, but provided hints for disease phenomena that would have otherwise remained obscure. 

### 4.1. Effect of P-Deprivation on Blood Pi

In cattle, anorexia and consumption of diets with inadequate P content are generally associated with a decline of plasma [Pi] [11,27]. In both studies presented here the cows received a ration approximately 40% below currently recommended daily dietary P supply for the dry period [28]. As expected, we observed a significant decline of plasma Pi concentrations in both LP and LPAP cows during dietary P depletion [24]. Interestingly, the nadir of ante-partum plasma [Pi] was lower in the first study compared to the second, despite the similarly low dietary P content. Part of the observed difference between experiments might be attributable to differences in feeding procedures. Whereas in Study I cows were fed twice daily, in Study II feed was offered through electronic feeding gates distributing feed consumption more evenly over the day. 

Lactating dairy cows are more prone to develop hypophosphatemia, due to the P excretion in milk [27,29]. This effect is of particular relevance in fresh cows, due to the decrease in feed intake around parturition and the sudden increase of P requirements for milk production [27,29]. This was demonstrated by the AP (control) group, which also developed hypophosphatemia in early lactation, despite receiving adequate P supplementation in the diet. Dietary P content before calving also affects P balance after calving. Dry cow diets low in P stimulate counter regulations which render cows faster in responding to the sudden loss of electrolytes after calving [21,24]. 

Continuous restricted dietary P supply after parturition (LP treatment) further exacerbated hypophosphatemia, with values below detection limit in some cows (<0.2 mmol/L). Blood Pi values restored promptly when cows were switched to adequate P diets.

### 4.2. Hemolysis and PPH and PPH 

Intravascular breakdown of RBCs results in hemoglobinemia [30]. When the concentration of free hemoglobin in the plasma surpasses the renal threshold for reabsorption, the urine acquires a reddish color, identifying hemoglobinuria [31]. In five of the 18 LP cows, intravascular hemolysis occurred to such an extent that brown to black urine was passed (LP-PPH2). This severe loss of RBC was clearly reflected by their low RBC counts and PCV in early lactation. LP-PPH1 cows did not show the typical discoloration of the urine, although their bloodwork indicated they were anemic. We believe that either the degree, or the location, of the hemolysis may provide an explanation for the differences between these two groups. It is possible that the degree of intravascular hemolysis in the LP-PPH1 group was not sufficient to exceed the renal threshold for hemoglobin. On the other hand, when breakdown of (abnormal) RBCs occurs, mainly in the spleen and the liver, anemia can be present without concomitant hemoglobinemia. In either scenario, an increase in TBil is expected, as bilirubin is formed by the breakdown of hemoglobin.

Across all treatments we found increased circulating TBil after calving. The effect of LN on TBil is not surprising, since total bilirubin in blood increases with advancing age of dairy cows [32,33]. Increases in plasma total bilirubin are usually ascribed to reduced biliary excretion of bilirubin from the blood. This is attributed to hampered conversion by the liver due to increased liver lipid accumulation in early lactation [34,35]. The course and height of the bilirubin concentrations were, however, very different for the clinically healthy and hematologically-affected animals. The subset of cows that did not show evidence of hemolysis, based upon their RBC count and PCV (AP, LP-PPH0) also did not display hyperbilirubinemia. In contrast, surges in TBil were observed in both LP-PPH1 and LP-PPH2 cows within the first week after parturition and were considerably more pronounced during the clinical phase of the disease. The TBIL values stratified by PPH status corroborated our assumption that hemolysis was likely to have started shortly after calving and, thus, before clinical signs became apparent and also occurred in the LP-PPH1 cows. 

The correlation analyses of both studies also showed a negative association of TBil with RBC was already present before hemolysis became apparent. This indeed suggested that some level of hemolysis might occur after calving.

### 4.3. Regenerative Anemia

The observed decline in PCV after calving in both control groups is consistent with findings of previous studies [14,15,16]. 

In all LP-PPH2 cows, MCV and MCH were still within normal ranges after parturition (ST3), but a continuous rise of MCV, that coincided with an increase in MCH, could be observed from the moment hemoglobinuria was first noted. The determined concentration of Hb in RBC by the ADVIA 2120i is not confounded by the presence of free Hb in plasma [36]. Blood smears of cows during the clinical phase of the disease showed evidence for intensified erythropoiesis, reflected in the presence of macrocytes, polychromasia, basophilic stippling, reticulocytes and nucleated RBCs. Anemia probably preceded clinical signs by a few days, since a certain time lag exists before signs of a bone marrow response become apparent [37]. The newly produced cells can be twice the normal RBC volume and contain more hemoglobin than adult RBCs [38,39,40]. The rise in MCV, and MCH in the LP-PPH1 and LP-PPH2 groups after ST3 might, thus, at least in part, be explained by (ongoing) RBC regeneration. The PCV and MCV results of the LP-PPH subgroups, confirmed that the degree of macrocytosis was related to the severity of the anemia [38,41,42]. The higher overall MCV and MCH during the second replicate of Study I (significant replicate effect) could be explained by the uneven distribution of LP-PPH1 and LP-PPH2 cows over the two replicates (more animals were affected in replicate 2). 

In conclusion, RBC indices of the LP-PPH1 and LP-PPH2 cows followed the course of regenerative anemia. In the LPAP, APAP, LP-PPH0 and AP cows no evidence for such an erythropoietic response could be detected in the blood. The high proportion of subclinically-affected cows also showed that the incidence of PPH might be greater than actually reported [5,43].

### 4.4. Linking Hypophosphatemia and PPH

A link between low plasma [Pi] concentrations and RBC destruction has been postulated before [2,3,6]. A decline of adenosine triphosphate (ATP) in the RBCs, caused by reduced availability of intracellular Pi (RBC [Pi]) has been proposed as the most important causative factor [7]. The RBC [Pi] reflects a double equilibrium: on one hand, the equilibrium between RBC [Pi] and plasma [Pi] and, on the other hand, the equilibrium between RBC [Pi] and intracellular organic phosphates [44]. How these equilibria are affected by hypophosphatemia is not completely clarified. 

Grünberg et al., studied dietary P deprivation in mid-lactating dairy cows, which resulted in low plasma [Pi] values (nadir of 0.55 ± 0.16 mmol/L) [11]. They found that, in contrast to the plasma [Pi], the RBC [Pi] was not significantly altered. [11]. This constant RBC [Pi] with declining P supply and hypophosphatemia, that was also observed for other tissues, such as muscle and liver, indicates that RBCs are highly effective in maintaining physiologic intracellular [Pi]. In these cows, RBC parameters were also not affected, despite a marked decline of plasma [Pi] for 5 weeks [11]. RBC parameters of the cows of study II were also not negatively affected. Whereas mid-lactation and ante-partum P deprivation alone did not seem to negatively affect the erythron, we found that extending dietary P deprivation into early lactation, apart from showing negative effects on DMI and milk production had a marked effect on the RBCs of some cows. Although all LP cows underwent the same degree and duration of hypophosphatemia, cows severely suffering from the consequences of PPH (LP-PPH2), and subclinically-affected (LP-PPH1) cows, as well as hematologically-unaffected cows (LP-PPH0), were present within this treatment group. 

We analyzed total P content of RBCs, since we hypothesized that cells eventually deplete both intracellular Pi and organic phosphates in times of (severe) hypophosphatemia. The numerical differences indeed suggested this may have happened in LP-PPH1 and LP-PPH2 cows. We measured the lowest RBC P content in cow 3739 10 d p.p. during the clinical phase of the disease. 

Hemolysis was not halted by oral P treatment in this cow, although administration of P salts resulted in prompt correction of the hypophosphatemia. PCV kept declining and osmotic resistance did not improve with oral P supplementation. Blood transfusion was necessary to assure the cow’s survival. In contrast, two LP PPH2 cows remained untreated and recovered without intervention and while remaining on the P deficient diet. The fact that the erythrocyte count of cow 8183 did not increase between day 18 and 24 relative to parturition, even though intensified erythropoiesis was evident, indicates that hemolysis was also not completely stopped, and, rather, the production of new erythrocytes was in equilibrium with an ongoing loss. Some authors suggest that irreversible changes occur in bovine RBCs during P deficiency, which might explain the irresponsive nature of PPH to P supplementation and the continuous breakdown of the RBCs [6]. 

### 4.5. Osmotic Resistance of RBC during Hypophosphatemia and PPH

In the current study, the comparison of the osmotic resistance between AP and LP cows revealed that RBCs of the latter group were less resistant to osmotic stress. It was, however, remarkable that this effect existed even during the acclimation period, and, thus, before dietary P deprivation was initiated. We have no explanation for this difference between the AP and LP groups, that was consistent over time throughout the entire study period. The numerically higher Na concentrations at which the RBCs of the LP-PPH2 cows lysed prior to clinical disease (compared to other groups) seem to point to an inherent mildly increased osmotic fragility. The biological relevance of the difference, however, remains uncertain. 

The osmotic resistance of AP, LP-PPH0 and LP-PPH1 cows did not change over time. The determination of osmotic resistance of RBC in cattle with clinical PPH in earlier studies has yielded conflicting results [1]. Ogawa et al. [6] found an increase in osmotic fragility in one cow, a few days before hemoglobinuria became apparent. Others found fragility tests in a herd with anemia and PPH within normal limits [45]. Based upon our results, one could conclude that RBC osmotic resistance in LP cows, in general, was not negatively affected over time, despite the experiencing of severe hypophosphatemia between ST2–ST4. We, however, gathered additional information from the samples taken during the hemolytic crisis. We determined that RBCs were less resistant to osmotic stress during the course of clinical PPH compared to the other sampling times. In the LP-PPH2 cows that survived the hemolytic crisis, an improvement of osmotic resistance was noted towards the end of the study. This could be explained by the increase in new RBCs (reticulocytes and young RBC) which are more osmotically resistant [6,46].

The question remains as to why certain fresh cows were more severely affected (LP-PPH2) than others (LP-PPH1) and why other cows were not affected at all (LP-PPH0), despite having similar low plasma Pi values (<0.3 mmol/L). It is widely believed that hypophosphatemia interferes with the glycolysis of RBCs and that the subsequent drop in intracellular ATP is the cause for RBC destruction [7]. Studies on the human RBC indicate that ATP must fall to less than 15% of normal before hemolysis occurs [47]. Although a strong relation exists between hypophosphatemia and RBC ATP concentrations, a fall of ATP to less than 15% of normal values does not happen rapidly. In a human case of hypophosphatemia, a plasma [Pi] concentration below 0.32 mmol/L for eight consecutive days was not sufficient to cause hemolysis even though ATP in the cells was decreased [48]. These results were in line with previous work in mid-lactation cows [11]. In vitro studies on bovine RBC, incubated for 72 h without added Pi, also did not demonstrate significant changes in osmotic fragility, deformability or cell morphology [7]. Unfortunately, ATP content of RBC was not measured in our study. Based on our correlation analysis, an association between plasma Pi and osmotic resistance seemed to exist, as well as an association between total intracellular P in the RBCs and osmotic resistance. It is not clear which of the two parameters was more relevant for this association. 

ATP depletion causes the RBCs to adopt a spherical shape [49]. Spheres have a decreased surface to volume ratio, which is one of the prime determinants of osmotic fragility [50]. Spherocytes were observed in the differential blood smears of the two cows during the clinical phase of the disease. Combined data from these cows suggested a link between the presence of spherocytes in blood and a decreased osmotic resistance during PPH. The transition from the dry phase to lactation seems to be the only time period during which a point can be reached at which RBCs are no longer able to deal with low plasma [Pi].

### 4.6. Hypophosphatemia and RBC Intracellular Ion Alterations

A decrease in RBC ATP, coinciding with low blood inorganic phosphate concentrations, has been found, among a number of species: in cattle [7], mice [51] and humans [13]. Therefore, hypophosphatemia could (in)directly affect RBC electrolyte content when insufficient ATP is available for adequate proton pump function. Theoretically, the reduction in Na+/K+ -ATPase activity in RBC membranes of P-deficient animals would, therefore, result in a decrease of RBC potassium and increase of RBC sodium concentrations [52]. In the current study, however, such overt alterations in intracellular electrolyte content did not occur, in general, even though hypophosphatemia lasted for several weeks. Despite the relatively stable RBC electrolyte content throughout the present study, some interesting findings were obtained from the clinical cases. A clear decrease in RBC potassium and increase in RBC sodium concentrations occurred in one cow during the clinical phase of the disease, implying impaired function of the Na+/K+ -ATPase in the RBC membranes. Admittedly, leakage of the RBC membranes could emanate a similar effect. Ogawa et al., also found a decrease (21.6 mmol/L a.p. to 15.7 mmol/L p.p.) in RBC potassium in a cow suffering from PPH [6]. In another LP-PPH2 cow (8183), although more subtle, such alterations were also noticed at the time when dark urine was passed. The increase of both RBC–K and RBC- = −Na after the hemolytic crisis in that cow could be explained by the increase in new RBCs (reticulocytes and young RBC) which have higher Na+/K+ -ATPase activity [6,46]. Ideally, RBCs would have been studied at a more frequent interval in the first 14 d after calving to more accurately study whether intracellular electrolyte content alterations precede, or only coincide with, PPH. 

### 4.7. Associated Risk Factors

From the current study it can be concluded that restricted dietary phosphorus supply throughout the periparturient period results in PPH, clinically and subclinically, after parturition in certain cows. The timing of the disease, in the second to fourth week of lactation, corresponds with that of previous reports [1,2,3]. A multitude of risk factors, such as parity, anorexia, high milk production and copper deficiencies, have been associated with the development of PPH. Herein, clinically-affected cows were in third or higher lactation. Nevertheless, high milk yield could not serve as a predictor for the development of PPH. DMI in the dry period also did not differ between treatment and PPH-subgroups. The fact that feed intake was higher in LP-PPH1 and LP-PPH2 cows immediately after calving (ST3), compared to LP-PPH0, would suggest that these subgroups were not more deprived of P by ingesting less P at that time. Nevertheless, after the first week of lactation, all P-deficient cows became anorectic and, accordingly, produced less milk [20].

Copper deficiency interferes with the antioxidant systems of the RBC, which are necessary to protect hemoglobin from oxidative denaturation, and this can result in anemia [53]. Liver Cu content is the most accurate parameter to assess Cu status of an animal [23], and concentrations below 20 mg of Cu/kg DM liver are indicative of a deficiency [54]. Therefore, we were able to rule out Cu deficiency with certainty. Since we also did not find differences in liver Cu content between PPH groups, we believe there was no interplay of copper status and PPH in the current study. 

In the first study, adequate P content of the AP cow diet was obtained by supplementing the diet with NaH_2_PO_4_. Subsequently, the sodium intake of AP cows was higher than sodium intake of LP cows. The measured dietary Na content of the lactation LP was in a marginal range for lactating dairy cows; 1.5 g/kg DM [20]. To further study the effect of this discrepancy in the diet on the occurrence of PPH, sodium and creatinine excretion in urine were analyzed. As expected, the Na:creatinine ratio was lower in the urine of LP cows. There were, however, no differences between the PPH-subgroups, and the nadir of urine Na:creatinine ratio in LP animals was reached after clinical PPH occurred. Although there is some evidence that RBC sodium concentrations can decrease when sodium intake is reduced for an extended period, hyponatremia does not affect RBC ATP concentrations [55]. Hyponatremia is associated with increased osmotic resistance, rather than a decrease [56].

Multiple studies have shown a gradual decrease in enzymatic activities with the aging of erythrocytes [57]. There is a decline in the levels of phosphate esters with aging of red cells [58] and the ability of erythrocytes to respond to metabolic stimulation (high plasma [Pi]) for glycolysis falls exponentially with increasing age and is, basically, lost in the oldest cells [59]. This suggests that the oldest cells, in particular, are sensitive to the low [Pi] in the plasma and would be the first cells to lyse. Interestingly, some researchers found a link between hypophosphatemia and accelerated erythrocyte aging. In mice, moderate phosphate depletion for four days significantly increased RBC susceptibility for apoptosis and phagocytic clearance [51]. They postulate that this premature ‘apoptotic’ erythrocyte death serves to eliminate defective erythrocytes from the blood. Rather than a one-on-one relation, we deem it more likely that hypophosphatemia, over time, - causes intrinsic alterations in the RBCs, rendering them more susceptible to both extravascular and intravascular hemolysis. The development of clinically-relevant anemia depends on the number and severity of the eryptosis [60]. 

## 5. Conclusions

Hypophosphatemia was successfully induced in cows receiving a P-deficient diet in both the ante- and post-partum period (LP treatment). Feeding a ration at least 40% below the daily P requirements solely throughout the dry period (LPAP; 0.15% DM) did not affect RBC counts, total bilirubin, MCV, MCH or milk yield throughout the study. PCV and RBC count decreased in all cows after parturition, but anemia and PPH developed only in cows that were phosphorus-depleted through both the ante- and post-partum periods. A moderate increase in circulating bilirubin was present in all cows following parturition, but hyperbilirubinemia was found in cows that were subclinically and clinically affected by PPH. RBC indices of affected cows followed the course of typical regenerative anemia. In cows receiving adequate phosphorus supplementation, no evidence for an erythropoietic response could be retrieved in the blood.

During the clinical phase of PPH, evidence of a decrease in RBC osmotic resistance and altered intracellular ion content was found. Based upon our results we reason that hypophosphatemia, over time, causes intrinsic alterations in the RBCs, rendering them more susceptible to hemolysis.

## Figures and Tables

**Figure 1 animals-13-00404-f001:**
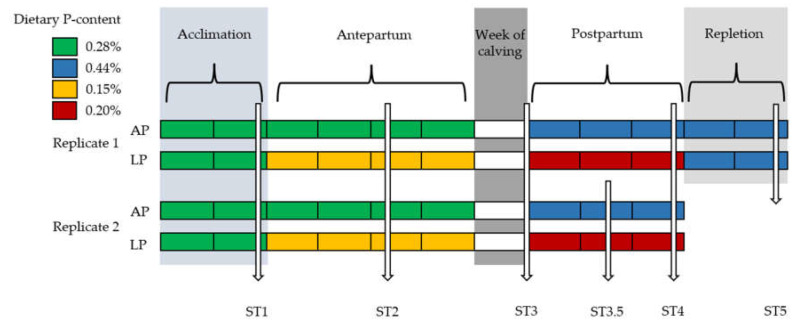
Experimental timeline for adequate-phosphorus (AP; n = 18) and low-phosphorus (LP; n = 18) treatments. Each rectangle represents one week. P content on a dry matter basis can be found in the legend. Week of calving is shown in white. During acclimation, cows in both treatments received the same dry cow diet with adequate P supply. Vertical arrows represent the sampling times at which the erythron and liver were studied.

**Figure 2 animals-13-00404-f002:**
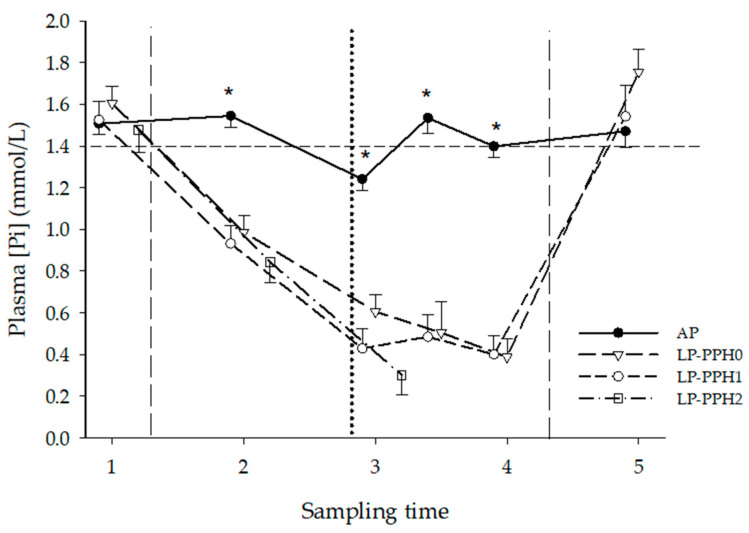
Plasma Pi concentration of dairy cattle receiving either a diet with adequate P content (AP, n = 18) or a P deficient diet (LP, n = 18) in the periparturient period. LP-PPH2: cows that developed clinical PPH (i.e., hemoglobinuria; n = 5), LP-PPH1: cows with subclinical PPH (n = 6); LP-PPH0: cows without evidence for PPH (n = 7). The vertical dashed lines mark the beginning and end of the phosphate depletion and repletion phase of the study. The vertical dotted line indicates calving. The horizontal dashed line presents the lower limit of the reference range for Plasma Pi in cattle. * means different from all other groups at that time.

**Figure 3 animals-13-00404-f003:**
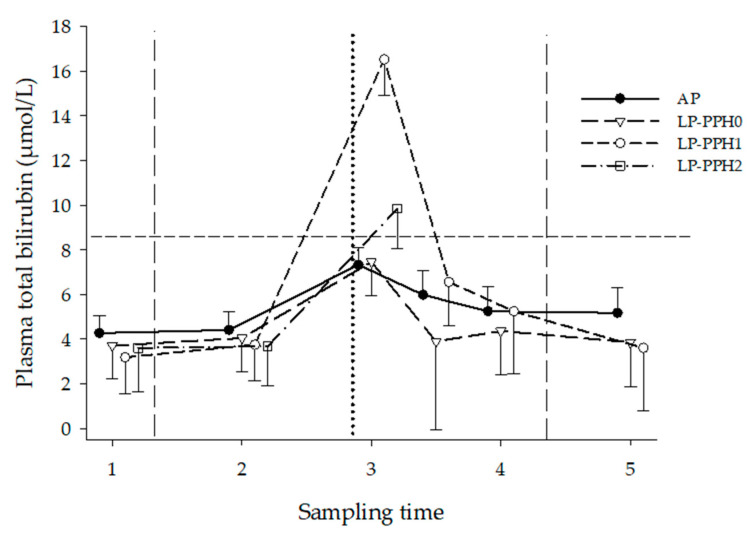
Plasma total bilirubin of dairy cattle receiving either a diet with adequate P content (AP, n = 18) or a P deficient diet (LP, n = 18) in the periparturient period. LP-PPH2: cows that developed clinical PPH (i.e., hemoglobinuria; n = 5), LP-PPH1: cows with subclinical PPH (n = 6); LP-PPH0: cows without evidence for PPH (n = 7). The vertical dashed lines mark the beginning and end of the phosphate depletion and repletion phase of the study. The vertical dotted line indicates calving. The horizontal dashed line presents the upper limit of the reference range for plasma bilirubin in cattle.

**Figure 4 animals-13-00404-f004:**
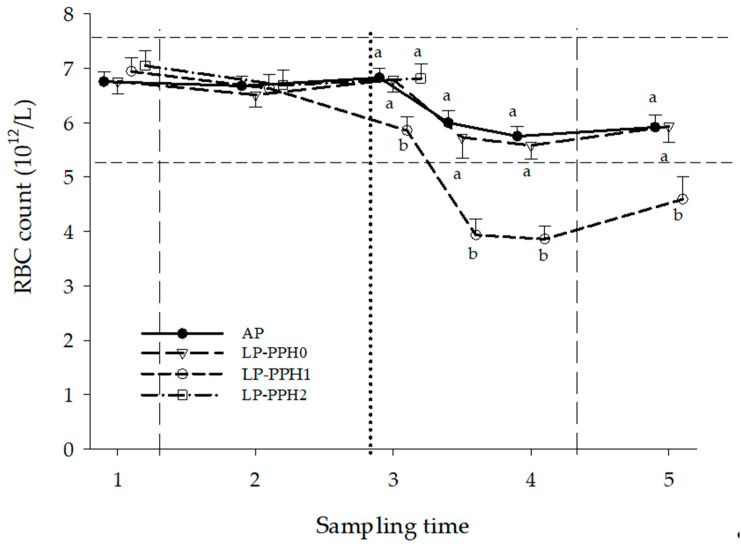
Red blood cell (RBC) counts of dairy cattle receiving a diet with adequate P content (AP, n = 18) or a P deficient diet (LP, n = 18) in the periparturient period. LP-PPH2: cows that developed clinical PPH (i.e., hemoglobinuria; n = 5), LP-PPH1: cows with subclinical PPH (n = 6); LP-PPH0: cows without evidence for PPH (n = 7). The vertical dashed lines mark the beginning and end of the phosphate depletion and repletion phase of the study. The vertical dotted line indicates calving. The horizontal dashed lines present the upper and lower limit of the reference range for RBC in cattle. Values of groups with different lower case letterss differ from each other at that specific sampling time.

**Figure 5 animals-13-00404-f005:**
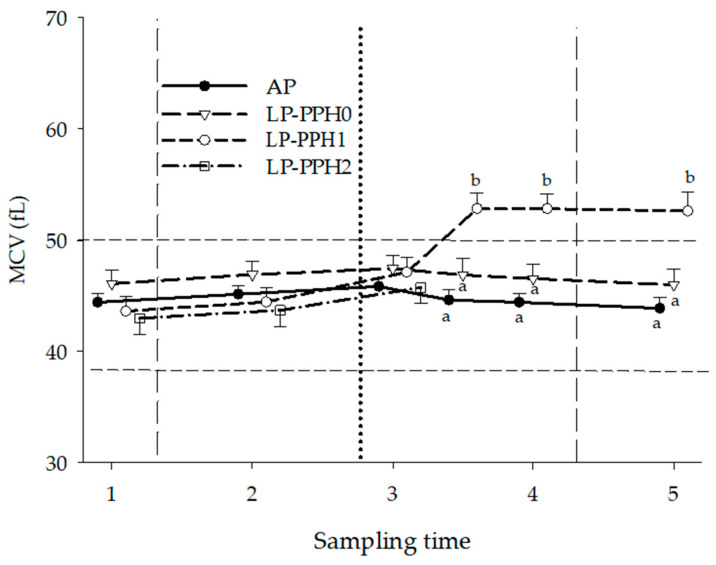
Mean corpuscular volume (MCV) of dairy cattle receiving a diet with adequate P content (AP, n = 18) or a P deficient diet (LP, n = 18) in the periparturient period. LP-PPH2: cows that developed clinical PPH (i.e., hemoglobinuria; n = 5), LP-PPH1: cows with subclinical PPH (n = 6); LP-PPH0: cows without evidence for PPH (n = 7). The vertical dashed lines mark the beginning and end of the phosphate depletion and repletion phase of the study. The vertical dotted line indicates calving. The horizontal dashed line presents the upper and lower limit of the reference range for MCV in cattle. Values with different superscripts differ from each other at that specific sampling time.

**Figure 6 animals-13-00404-f006:**
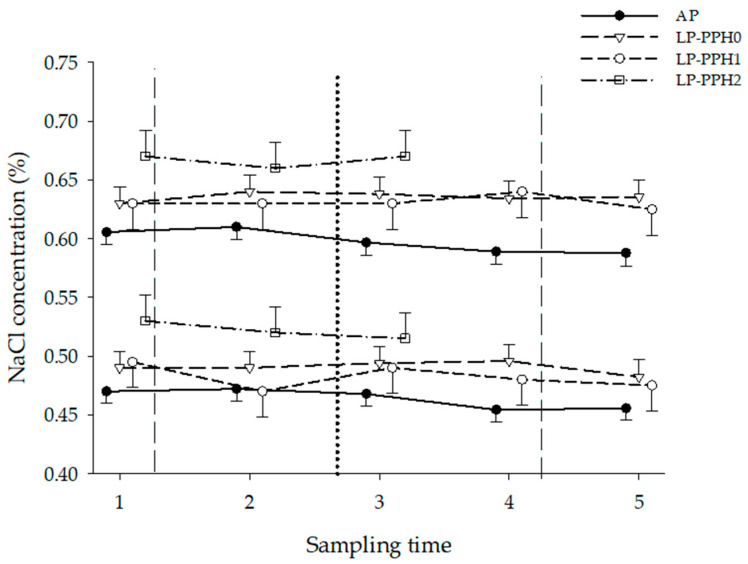
Osmotic resistance of dairy cattle receiving a diet with adequate P content (AP, n = 9) or a P deficient diet (LP, n = 9) in the periparturient period. Concentrations of NaCl solutions at which 10% (upper panel) and 90% (lower panel) of hemolysis occurred at the different blood sampling times. LP-PPH2: cows that developed clinical PPH (i.e., hemoglobinuria; n = 2), LP-PPH1: cows with subclinical PPH (n = 2); LP-PPH0: cows without evidence for PPH (n = 5). The vertical dashed lines mark the beginning and end of the phosphate depletion and repletion phase of the study. The vertical dotted line indicates calving.

**Figure 7 animals-13-00404-f007:**
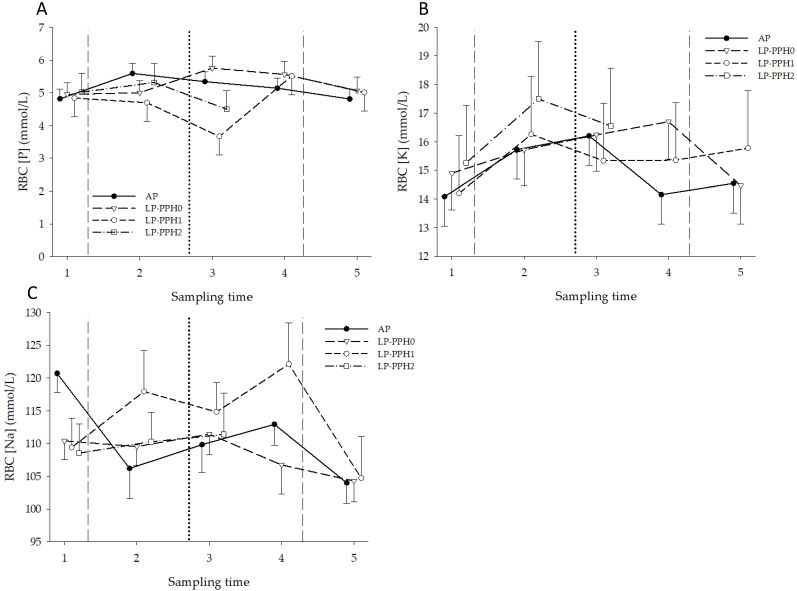
RBC intracellular phosphate (**A**), potassium (**B**), and sodium (**C**) concentrations in the periparturient period of dairy cattle receiving a diet with adequate P content (AP) or a P deficient diet (LP) in the periparturient period. LP-PPH2: cows that developed clinical PPH (i.e., hemoglobinuria; n = 2), LP-PPH1: cows with subclinical PPH (n = 2); LP-PPH0: cows without evidence for PPH (n = 5). The vertical dashed lines mark the beginning and end of the phosphate depletion and repletion phase of the study. The vertical dotted line indicates calving.

**Figure 8 animals-13-00404-f008:**
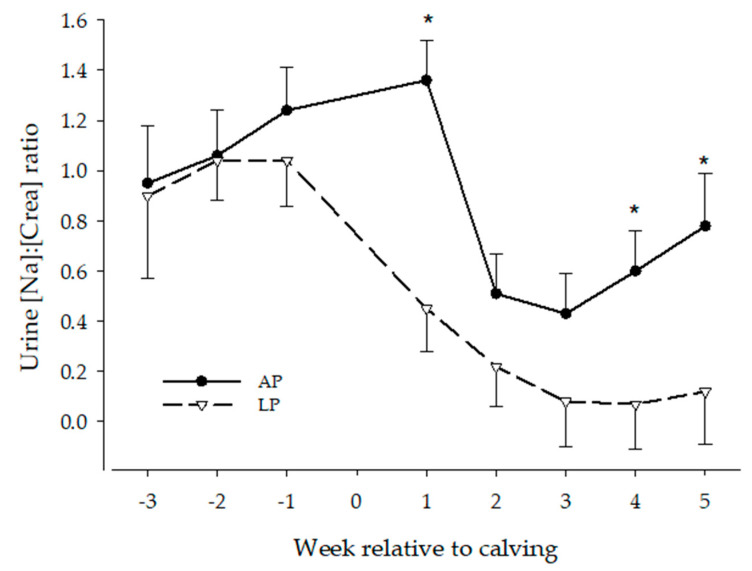
Urine Na:creatinine ratio of dairy cattle receiving either a P deficient diet (LP, n = 9) or a diet with adequate P content (AP, n = 9) in the periparturient period. * means significant different from other treatment at that time.

**Figure 9 animals-13-00404-f009:**
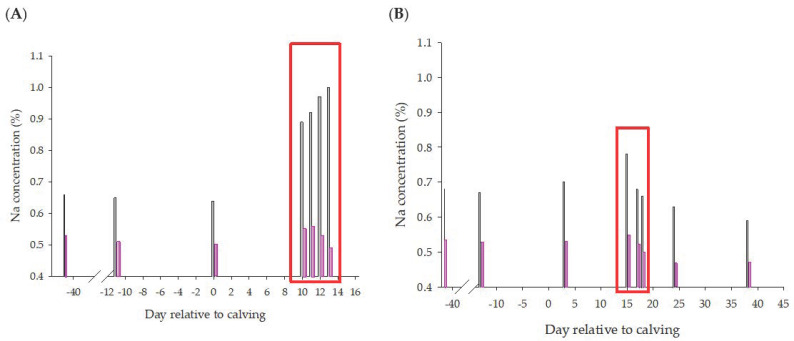
Osmotic resistance of two cows with periparturient hemoglobinuria (LP-PPH2, 3739 (**A**) and 8183 (**B**)). Concentrations of NaCl solution at which 10% (open) and 90% (purple bar) of hemolysis was induced throughout the P deprivation study. Red rectangle indicated timing of clinical disease.

**Figure 10 animals-13-00404-f010:**
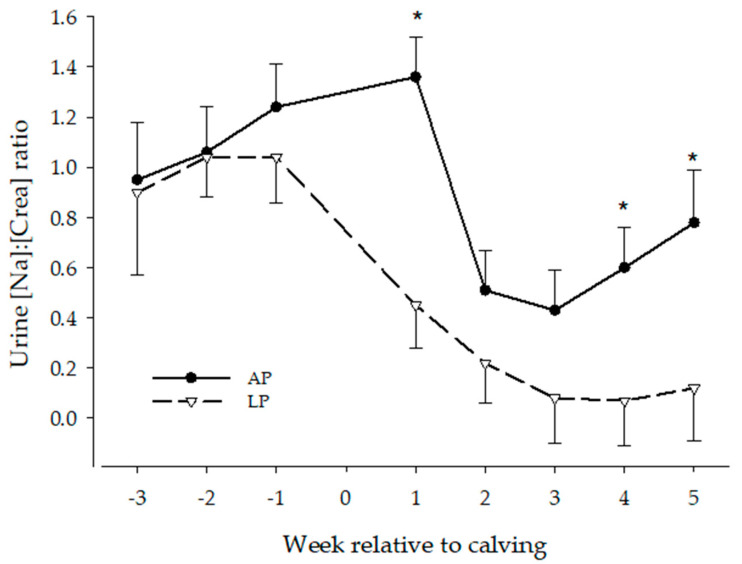
Packed cell volume of dairy cattle receiving either a P deficient diet (LP, n = 15) or a diet with adequate P content (AP, n = 15) in the dry period.

**Table 1 animals-13-00404-t001:** Several parameters of LP-PPH2 cows after parturition.

Parameter ^1^										
Drel ^2^	PCV	RBC	Plasma Pi	TBil	MCV	Urine	P-RBC	K-RBC	Na-RBC	Diff.
RR	22–33	5.1–7.6	1.4–2.3	<8.6	38–50					
** *3739* **	Prematurely released									
0 ^3^	32	7.0	0.38	9.4	46.0		4.5	18.0	111	
7				36.9		normal				
10	17	3.5	0.44	44.1	49.5	dark brown	3.1	11.3		SC
11	14	2.8	0.56		50.5	black	4.0	10.3		SC
12	13	2.4			52.0	brown/red	4.5	9.5	142	SC, BS
13	9	1.7	2.74		54.5	brown/red	5.6	13.3	161	SC, BS, 4.9% RetiAbs
** *8183* **	Completed study									
3 ^3^	32	7.3	0.35	12.0	43.9		4.5	15.1		
12				11.8		normal				
14			0.22			brown				
15	18	3.6	0.21		50.5	dark brown	5.5	12.9	118	several SC, BS
17	15	2.8	0.18		53.4	brown	6.5	16.9	97	some SC, BS
18	15	2.8	0.21		55.8	normal	4.7	15.1	104	no SC, BS, metarubricytes
24 ^4^	18	2.8	0.68	2.8	65.0		5.5	27.8	149	
38 ^5^	25	4.3	1.54	4.0	58.3		6.1	26.2	125	
	PCV	RBC	Plasma Pi	TBil	MCV	Urine			RetiAbs	
** *1261* **	Prematurely released									
4 ^3^	29	5.9	0.2	9.2	49.1				1.7	
9			0.17			normal				
11	16	3.0	0.32	22.2	52.4	dark, not transparent				
13	12	2.2			57.5	black, not transparent				
** *3340* **	Prematurely released									
3 ^3^	30	6.0	0.19	8.1	50.7				2.6	
8			0.22			normal				
10	19	3.6	0.37	23.5	53.3	dark brown				
12	11	1.8	0.42		58.4	dark, not transparent				
** *5035* **	Completed study									
1 ^3^	32	7.6	0.41	10.5	42.4				5	
8			0.33	27.7		normal				
11	18	3.9	0.51		46.3	brown				
13 ^3.5^	12	2.6	0.42		48.2	transparent, slightly brown			7.5	
15			0.56	25.4		normal				
22 ^4^	17	2.8	0.45	14	60.3				7.5	

^1^ PCV = packed cell volume (%), RBC = Red blood cell (×10^12^/L), Plasma Pi (mmol/L), TBil = Total bilirubin (μmol/L), MCV = Mean corpuscular volume (fL), P-RBC = RBC intracellular P (mmol/L), K-RBC = RBC intracellular potassium (mmol/L), Na-RBC = RBC intracellular (mmol/L), Diff. = differential blood count, SC: spherocytes, BS: basophilic stippling, RetiAbs = absolute number of reticulocytes (×10^9^/L). RR = reference range for each parameter. ^2^ Drel = Days relative to parturition. ^3^ Corresponds with ST3 of the study. ^3.5^ Corresponds with ST3.5 of the study. ^4^ Corresponds with ST4 of the study.

**Table 2 animals-13-00404-t002:** Packed cell volume (%) of periparturient dairy cattle receiving AP or LP diet.

	Sampling Time						*p* Value		
Treatment ^1^	ST1	ST2	ST3	ST3.5	ST4	ST5	Treatment	Time	T × T
AP	29.9 ± 0.7 ^a^	30.1 ± 0.7 ^a^	31.1 ± 0.6 ^a^	26.9 ± 0.8 ^b^	25.6 ± 0.6 ^b^	25.4 ± 0.9 ^b^	0.01	<0.0001	0.002
LP	30.5 ± 0.7 ^a^	29.6 ± 0.7 ^a^	30.0 ± 0.6 ^a^	21.6 ± 0.9 ^c,^*	22.4 ± 0.7 ^c,^*	25.0 ± 1.0 ^b^			
PPH-group ^2^	ST1	ST2	ST3	ST3.5	ST4	ST5	Group	Time	G x T
LP-PPH0 LP-PPH1 LP-PPH2	30.9 ± 0.9 ^a^	30.3 ± 0.9 ^a^	31.8 ± 0.9 ^a^	27.3 ± 1.6 ^a,b^	26.0 ± 0.9 ^b,^*	26.3 ± 1.1 ^b^	<0.0001	<0.0001	0.0002
	30.3 ± 0.9 ^a^	29.3 ± 0.9 ^a^	27.2 ± 0.9 ^a,b,^*	20.1 ± 1.1 ^c,^*	20.4 ± 0.9 ^c^	23.2 ± 1.6 ^b,c^			
	30.2 ± 1.0 ^a^	29.0 ± 1.0 ^a^	31.0 ± 1.0 ^a^	(11.9 ± 2.2) ^c,^*	(17.4 ± 1.6) ^c^	(24.9 ± 2.3) ^b^			

^a–c^ Mean values in the same row with different superscripts differ (*p* < 0.05 Bonferroni corrected) from each other. Mean values in the same column with an asterisk (*) differ (*p* < 0.05 Bonferroni corrected, if appropriate) from the other treatment or groups. ^1^ AP = diet with adequate amount of phosphorus. LP = diet with low phosphorus content. ^2^ PPH = Postparturient hemoglobinuria. LP-PPH0: cows without evidence for PPH (n = 7), LP-PPH1: cows with subclinical PPH (n = 6); LP-PPH2: cows with clinical PPH (n = 5). Data of LP-PPH2 cows presented in parenthesis only include animals that remained in the study after clinical disease (n = 2).

**Table 3 animals-13-00404-t003:** Mean corpuscular hemoglobin (fmol) of periparturient dairy cattle receiving AP or LP diet.

	Sampling Time							*p* Value	
Treatment ^1^	ST1	ST2	ST3	ST3.5	ST4	ST5	Treatment	Time	T × T
AP	1.02 ± 0.02 ^a^	1.05 ± 0.02 ^a^	1.05 ± 0.02 ^a^	1.04 ± 0.02 ^a^	1.03 ± 0.02 ^a^	1.03 ± 0.02 ^a^	0.003	<0.0001	<0.0001
LP	1.01 ± 0.02 ^c^	1.05 ± 0.02 ^b^	1.07 ± 0.02 ^b^	1.14 ± 0.02 ^a,^*	1.17 ± 0.02 ^a,^*	1.14 ± 0.02 ^a,^*			
PPH-group ^2^	ST1	ST2	ST3	ST3.5	ST4	ST5	Group	Time	G x T
LP-PPH0 LP-PPH1 LP-PPH2	1.04 ± 0.03 ^a^	1.08 ± 0.03 ^a^	1.08 ± 0.03 ^a^	1.08 ± 0.03 ^a^	1.07 ± 0.03 ^a^	1.06 ± 0.03 ^a,^*	NS	<0.0001	<0.0001
	1.00 ± 0.03 ^b^	1.03 ± 0.03 ^b^	1.06 ± 0.03 ^b^	1.17 ± 0.03 ^a^	1.19 ± 0.03 ^a,^*	1.20 ± 0.04 ^a^			
	0.99 ± 0.03 ^c^	1.02 ± 0.03 ^c^	1.04 ± 0.03 ^c^	(1.18 ± 0.05) ^b^	(1.41 ± 0.04) ^a,^*	(1.30 ± 0.05) ^b^			

^a–c^ Mean values in the same row with different superscripts differ (*p* < 0.05 Bonferroni corrected) from each other. Mean values in the same column with an asterisk (*) differ (*p* < 0.05 Bonferroni corrected, if appropriate) from the other treatment or PPH-groups.^1^ AP = diet with adequate amount of phosphorus, n = 18. LP = diet with low phosphorus content, n = 18. ^2^ PPH = Postparturient hemoglobinuria. LP-PPH0: cows without evidence for PPH (n = 7), LP-PPH1: cows with subclinical PPH (n = 6); LP-PPH2: cows with clinical PPH (n = 5). Data of LP-PPH2 cows presented in parenthesis only include animals that remained in the study after clinical disease (n = 2).

**Table 4 animals-13-00404-t004:** Liver Cu content (mg/kg DM) in the liver of periparturient dairy cattle receiving AP or LP diet.

	Sampling Time					*p* Value		
Treatment ^1^	ST1	ST2	ST3	ST4	ST5	Treatment	Time	T × T
AP	357 ^a^ (327–390)	371 ^a^ (340–406)	384 ^a^ (351–419)	286 ^b^ (262–313)	362 ^a^ (329–399)	NS	<0.0001	0.0005
LP	294 ^a^ (269–321)	301 ^a^ (276–329)	314 ^a^ (287–343)	292 ^a^ (266–321)	264 ^a,^* (236–294)			

^a,b^ Median and 95% CI of liver Cu content. Values in the same row with different superscripts differ (*p* <0.05 Bonferroni corrected) from each other. Values in the same column with an asterisk (*) differ (*p* < 0.05) from the other treatment. ^1^ AP = diet with adequate amount of phosphorus, n = 18. LP = diet with low phosphorus content, n = 18.

**Table 5 animals-13-00404-t005:** Effect of antepartum phosphorus deprivation on RBC count, PCV, MCV and MCH in periparturient dairy cows.

	Parameter ^1^											
	**RBC (10^12^/L)**			**PCV (%)**			**MCV (fl)**			**MCH (fmol)**		
	5.1–7.6			22–33			38–50			0.87–1.12		
**Treatment**	NS			NS			NS			NS		
**Time**	<0.0001			<0.0001			<0.0001			NS		
T × T	NS			NS			NS			NS		
Wrel ^2^	APAP		LPAP	APAP		LPAP	APAP		LPAP	APAP		LPAP
−6	6.3 ± 0.4		6.6 ± 0.3	29.4 ± 1.7		30.3 ± 1.7	50.4 ± 1.5		47.7 ± 1.5	1.03 ± 0.03		1.01 ± 0.03
−5	7.0 ± 0.2		6.8 ± 0.2	32.6 ± 1.0		30.4 ± 0.9	50.0 ± 1.1	♦	48.6 ± 1.1	1.07 ± 0.02		1.01 ± 0.02
−4	6.7 ± 0.1		6.8 ± 0.1	30.9 ± 0.6		30.6 ± 0.6	49.9 ± 1.0	♦	49.1 ± 1.0	1.04 ± 0.02		1.02 ± 0.01
−3	6.6 ± 0.1		6.8 ± 0.1	31.0 ± 0.6		31.2 ± 0.6	49.9 ± 1.0	♦	49.3 ± 1.0	1.05 ± 0.01		1.02 ± 0.01
−2	6.6 ± 0.1		6.8 ± 0.1	31.1 ± 0.5		30.8 ± 0.5	51.3 ± 1.0		49.9 ± 1.0	1.05 ± 0.01		1.02 ± 0.01
−1	6.6 ± 0.1		6.9 ± 0.1	31.1 ± 0.5		31.3 ± 0.5	51.5 ± 1.0		50.7 ± 1.0	1.05 ± 0.01		1.03 ± 0.01
1	7.0 ± 0.1	♦	6.9 ± 0.1	33.3 ± 0.5	♦	32.1 ± 0.5	52.0 ± 1.0		51.0 ± 1.0	1.06 ± 0.01		1.03 ± 0.01
2	6.3 ± 0.1	♦	6.5 ± 0.1	29.5 ± 0.5	♦	30.1 ± 0.5	51.0 ± 1.0		49.7 ± 1.0	1.05 ± 0.01		1.03 ± 0.01
3	6.0 ± 0.1	♦	6.3 ± 0.1	28.0 ± 0.5	♦	28.4 ± 0.5	50.1 ± 1.0	♦	49.0 ± 1.0	1.05 ± 0.01		1.02 ± 0.01
4	5.7 ± 0.1	♦	5.9 ± 0.1	26.7 ± 0.5	♦	27.1 ± 0.5	49.8 ± 1.0	♦	48.5 ± 1.0	1.04 ± 0.01		1.02 ± 0.01

^1^ RBC = red blood cell, PCV = Packed cell volume, MCV = Mean corpuscular volume, MCH = Mean corpuscular hemoglobin. Reference ranges for cattle provided for each parameter. ^2^ Wrel: week relative to parturition. Values with an ♦ differ significantly from the value at −1 week relative to parturition (*p* ≤ 0.05; Bonferroni corrected). APAP: adequate phosphorus antepartum. LPAP: low phosphorus antepartum.

## Data Availability

The data presented in this study are openly available in Researchgate at DOI: https://doi.org/10.13140/RG.2.2.35103.02723 (https://www.researchgate.net/publication/366408481_Datasheet_postparurient_hemoglobinuria, accessed on 15 December 2022).

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
