# Peer review of "Effect of Dietary Phosphate Deprivation on Red Blood Cell Parameters of Periparturient Dairy Cows"

_animals, 2023, doi:10.3390/ani13030404_

Round 1

Reviewer 1 Report

The paper is well written, the study well planned and topics is of interest. However, the manuscript is definitely too long: I suggest including only the clinical-pathological results avoiding other results (e.g feed intake, milk yield, the Cu liver content)

Please remove also all the part already published elsewhere: paragraph 3.2.1, 3.2.3 and the clinical cases that do not add anything significant to the manuscript, but only overburden the text.

Also the discussion is too long and unfocused.

Please remove from the discussion paragraph 4.7.

Lines 124: it’s not clear where these subset of cows coming from.

Lines 190-193: why only urine with visible haemoglobinuria were considered? Could be of interest also measured haemoglobin in urine from cow with a subclinical PPH (group LP-PPH1) to better define the extent and location of haemolysis ( if it was really extravascular as discussed by the authors)

Line 222: the content of P diet in the second study is different compare the first ( 0,15% vs to 0,16 for LP diet and 0,28 vs 0,30  for AP) why?

Table 1 Please, add reference interval for each parameters.

Fig 6: it’s not clear, please remove and substitute with one similar to the others.

Author Response

We appreciate this reviewers thorough review and thoughtful comments and suggestions.

The paper is well written, the study well planned and topics is of interest. However, the manuscript is definitely too long: I suggest including only the clinical-pathological results avoiding other results (e.g feed intake, milk yield, the Cu liver content)

Please remove also all the part already published elsewhere: paragraph 3.2.1, 3.2.3 and the clinical cases that do not add anything significant to the manuscript, but only overburden the text.

AU: We certainly agree with this reviewer's comment that the paper is long. 

Based on this comment we decided to delete sections that were not critically relevant for our conclusions. This includes data related to feed intake, milk yield and reticulocyte counts. We furthermore ended up deleted for examples paragraphs 3.2.1. and 3.2.3. We furthermore shortened several paragraphs of the discussion and deleted (numbering of chapters was evidently adjusted)

We deliberately chose not to remove the liver Cu data and the clinical cases, as we strongly believe that these findings were very important, as they help to answer our research questions and hypotheses.

You will see that our effort results in reduction of the paper volume by over 10 pages. We therefore deem to have duly complied with this reviewers request.

Also the discussion is too long and unfocused.

AU: the discussion was extensively revised and shortened. Some data of minor relevance were deleted;  werestructured part of the discussion to allow to present it in a more concise manner. The volume of the manuscript was reduced by over 10 pages.

Please remove from the discussion paragraph 4.7.

This paragraph was merged / shortened

Lines 124: it’s not clear where these subset of cows coming from.

AU: The sentence has been reworded for clarity

Lines 190-193: why only urine with visible haemoglobinuria were considered? Could be of interest also measured haemoglobin in urine from cow with a subclinical PPH (group LP-PPH1) to better define the extent and location of haemolysis ( if it was really extravascular as discussed by the authors)

AU: Thank you for this very valid question / comment. However when it comes to assessing the level of hemolysis  the appearance of hemoglobin of urine is only a very crude indicator. We included this parameter as it is the one this is typically associated with PPH. Little is known about the renal threshold for renal Hb- excretion and how much interindividual variation there is. With the intense monitoring of the PCV in study animals and the photometric screening for indication of hemolysis in plasma we deem to have had more precise diagnostic tools available. In our opinion measuring Hb in urine would not have increased the sensitivity of the detection of animals with subclinical intravascular hemolysis. 

Line 222: the content of P diet in the second study is different compare the first ( 0,15% vs to 0,16 for LP diet and 0,28 vs 0,30  for AP) why?

AU: rations were designed to contain the same P content, but determined percentages did differ slightly. Wording was adapted to clarify this.

Table 1 Please, add reference interval for each parameters.

AU: Very helpful addition to this table! Was adopted. Also in Table 6.

Fig 6: it’s not clear, please remove and substitute with one similar to the others.

AU: Done

Reviewer 2 Report

I was concernedc about the low nuber of animals/treatment. However, since a power test was used and the design was reviewed and acceoted in the associated Journsl of Dairy Science paper, I can accept the design.

This paper is of high importnce to the industry relativre to low diet P. It would be interesting to know if a 10-20% reductionn in diet P would have similar results.

Author Response

We thank this reviewer for the thoughtful and supportive comments